# Inflammation and Syndecan-4 Shedding from Cardiac Cells in Ischemic and Non-Ischemic Heart Disease

**DOI:** 10.3390/biomedicines11041066

**Published:** 2023-04-01

**Authors:** Mari E. Strand, Maarten Vanhaverbeke, Michiel T. H. M. Henkens, Maurits A. Sikking, Karoline B. Rypdal, Bjørn Braathen, Vibeke M. Almaas, Theis Tønnessen, Geir Christensen, Stephane Heymans, Ida G. Lunde

**Affiliations:** 1Institute for Experimental Medical Research, Oslo University Hospital and University of Oslo, 0450 Oslo, Norway; m.e.strand@medisin.uio.no (M.E.S.); theis.tonnessen@medisin.uio.no (T.T.); geir.christensen@medisin.uio.no (G.C.); i.g.lunde@medisin.uio.no (I.G.L.); 2Cardiology Department, AZ Delta, 8800 Roeselare, Belgium; vanhaverbeke.maarten@gmail.com; 3Netherlands Heart Institute (NLHI), 3511 EP Utrecht, The Netherlands; michiel.henkens@mumc.nl; 4Department of Pathology, CARIM, Maastricht University Medical Centre, 6229 HX Maastricht, The Netherlands; 5Department of Cardiology, CARIM, Maastricht University Medical Centre, 6229 HX Maastricht, The Netherlands; maurits.sikking@mumc.nl; 6Institute of Clinical Medicine, University of Oslo, 0315 Oslo, Norway; 7K.G. Jebsen Center for Cardiac Biomarkers, University of Oslo, 0315 Oslo, Norway; 8Division of Diagnostics and Technology, Akershus University Hospital, 1478 Lørenskog, Norway; 9Department of Cardiothoracic Surgery, Oslo University Hospital Ullevål, 0450 Oslo, Norway; braathen.bjorn@gmail.com; 10Department of Cardiology, Oslo University Hospital Rikshospitalet, 0372 Oslo, Norway; vibalm@ous-hf.no; 11Department of Cardiovascular Science, University of Leuven, 3000 Leuven, Belgium; stephane.heymans@mumc.nl

**Keywords:** proteoglycan, extracellular matrix, biomarker, heart failure, fibrosis, immunotherapy

## Abstract

Circulating biomarkers reflecting cardiac inflammation are needed to improve the diagnostics and guide the treatment of heart failure patients. The cardiac production and shedding of the transmembrane proteoglycan syndecan-4 is upregulated by innate immunity signaling pathways. Here, we investigated the potential of syndecan-4 as a blood biomarker of cardiac inflammation. Serum syndecan-4 was measured in patients with (i) non-ischemic, non-valvular dilated cardiomyopathy (DCM), with (*n* = 71) or without (*n* = 318) chronic inflammation; (ii) acute myocarditis (*n* = 15), acute pericarditis (*n* = 3) or acute perimyocarditis (23) and (iii) acute myocardial infarction (MI) at day 0, 3 and 30 (*n* = 119). Syndecan-4 was investigated in cultured cardiac myocytes and fibroblasts (*n* = 6–12) treated with the pro-inflammatory cytokines interleukin (IL)-1β and its inhibitor IL-1 receptor antagonist (IL-1Ra), or tumor necrosis factor (TNF)α and its specific inhibitor infliximab, an antibody used in treatment of autoimmune diseases. The levels of serum syndecan-4 were comparable in all subgroups of patients with chronic or acute cardiomyopathy, independent of inflammation. Post-MI, syndecan-4 levels were increased at day 3 and 30 vs. day 0. IL-1Ra attenuated IL-1β-induced syndecan-4 production and shedding *in vitro*, while infliximab had no effect. In conclusion, syndecan-4 shedding from cardiac myocytes and fibroblasts was attenuated by immunomodulatory therapy. Although its circulating levels were increased post-MI, syndecan-4 did not reflect cardiac inflammatory status in patients with heart disease.

## 1. Introduction

Heart failure is a common syndrome of multiple etiologies, carrying high morbidity, mortality and costs for society. Despite therapeutic advancement, the management of heart failure remains a significant challenge, and research efforts are increasingly focused on identifying novel cardiovascular biomarkers to aid in diagnosis, risk stratification, predict prognosis and guide treatment [1].

The innate immune system is activated in tissue injury, in response to hemodynamic stress or during infection. Inflammation is a central pathophysiological process across the entire spectrum of human or experimental heart failure, whether acquired or genetic, acute or chronic, and is associated with a poor outcome independent of etiology [2,3,4,5]. Nevertheless, trials attempting immunomodulating therapy have not been consistent, and although there is hope generated from certain trials [6,7], an improved mechanistic understanding of cardiac inflammation, finding new drug targets, developing drugs, and designing appropriate trials are needed, if anti-inflammatory treatment is to be implemented for heart failure [2,8]. To this end, circulating biomarkers reflecting cardiac inflammation are sought to provide clinicians with valuable information and to reduce the use of biopsies to describe the inflammatory status of the heart.

Syndecan-4 is a transmembrane proteoglycan that is part of an evolutionary old, four-membered family encoded by the *SDC1-4* genes [9,10,11,12]. They consist of a core protein substituted with three-five covalently attached heparan sulfate (HS) glycosaminoglycan (GAG) chains on the extracellular domain. The short cytoplasmic signaling domain, the transmembrane domain, the attachment sites for the GAGs and the extracellular cleavage domain are highly conserved. Syndecans have been implicated in a wide variety of functions in numerous cell types and tissues, and typically function as co-receptors. The MW of syndecan-4 is around 20 kDa, although it normally forms homodimers and is found in protein gels at around 35–40 kDa. The HS-substituted ectodomains can be enzymatically cleaved from cell surfaces and released into the local milieu and circulation [13]; termed shedding. Enzymes responsible for shedding are called sheddases and include matrix metalloproteinases (MMPs), thrombin and endopeptidases such as seralysins, astacins and a disintegrin and metallo-proteinases (ADAMs). 

Syndecan-4 is upregulated in hearts of patients and mice with pressure overload, myocardial infarction (MI), or infection [14,15,16,17,18,19]. Importantly, we have shown that mediators of innate immunity, i.e., the bacterial component lipopolysaccharide (LPS), tumor necrosis factor (TNF)α and interleukin (IL)-1β, upregulate the production and shedding of syndecan-4 from cardiac cells [18,20]. The shedding of syndecans creates soluble effectors that may act as reservoirs of chemotactic ligands and growth factors, thus promoting immune cell recruitment and wound healing [21]. We have demonstrated syndecan-4 shedding in diseased hearts, regulating the levels of infiltrating immune cells to the heart [18,20].

Several studies have investigated syndecan-1 [22,23,24,25,26,27,28] and -4 [29,30,31,32] as potential circulating biomarkers in heart disease, with a focus on ischemia and MI, dilated cardiomyopathy (DCM), and acute and chronic heart failure. We here hypothesized that the amounts of circulating syndecan-4 were increased in patients with inflammatory heart disease. To test our hypothesis, we used a commercially available enzyme-linked immunosorbent assay (ELISA) kit detecting syndecan-4, and recruited a relatively high number of patients for blood sampling with detailed phenotyping. This included separating the patients into inflammatory and non-inflammatory DCM based on the quantification of infiltrating cells through immunohistochemistry analysis of biopsies. Moreover, we hypothesized that immunomodulatory therapy would reduce syndecan-4 levels and shedding from cardiomyocytes and cardiac fibroblasts, and thereby reduce the pro-inflammatory syndecan-4-mediated effects locally in the heart. To test this, we virally overexpressed syndecan-4 in primary heart cultures and used a custom-made syndecan-4 antibody to readily detect syndecan-4 and syndecan-4 fragments. 

## 2. Materials and Methods

### 2.1. Serum Samples from Patients

Serum samples were obtained from *n* = 430 extensively phenotyped and genotyped cardiomyopathy patients within the Maastricht Cardiomyopathy Registry, Maastricht University Medical Center (MUMC+), Maastricht, the Netherlands [33]. The cohort consisted of DCM patients with or without chronic inflammation (*n* = 71 and 318, respectively) and subgroups of acute cardiomyopathies with specific diagnoses, i.e., myocarditis (*n* = 15), pericarditis (*n* = 3) and perimyocarditis (*n* = 23). The separation of patients into inflammatory (chronic myocarditis) and non-inflammatory DCM was based on the presence of inflammation defined as >14 infiltrating cells/mm^2^ on immunohistochemistry. Briefly, 4 μm formalin-fixed paraffin-embedded endomyocardial tissue sections were stained with H&E, Sirius red, CD3+, and CD45+ antibodies, as described previously [34]. 

Serum samples were obtained from *n* = 119 patients with myocardial infarction at university hospitals in Leuven, Belgium. In brief, patients presenting with MI, with or without ST-elevation, and scheduled for coronary angiography between October 2013 and April 2015 were prospectively enrolled and extensively phenotyped. Serum samples were obtained at hospital admission and at day 3 and day 30 post-MI. See [35] for the detailed method description, exclusion criteria and follow-up care of patients.

Serum samples were obtained during aortic valve replacement surgery at Oslo University Hospital Ullevål, Oslo, Norway, from *n* = 15 patients with severe, symptomatic aortic stenosis (AS). These patients constitute the subgroup receiving cold blood cardioplegia of a previously published cohort [36]. Blood was drawn from a coronary sinus catheter and from the cannulated radial artery immediately after onset of cardiopulmonary bypass and before clamping of the aorta. See [36] for detailed method description and patient characteristics.

### 2.2. Myocardial Tissue Samples from Patients

Interventricular myocardial biopsies were obtained during septal myectomy at Oslo University Hospital Rikshospitalet, Oslo, Norway, from *n* = 3 patients with hypertrophic obstructive cardiomyopathy (HOCM). Detailed protocol description and patient characteristics, including genetics, have been previously reported [37]. 

### 2.3. Cardiac Myocyte and Fibroblast Cultures from Neonatal Rats

Cardiac myocyte and fibroblast cultures were prepared from hearts of neonatal (1–3 days old) Wistar rats (Janvier Labs, Le Genest-Saint-Isle, France), as described [20]. Briefly, atrial tissue was removed, the ventricular tissue digested mechanically in collagenase/pancreatin, before transfer to uncoated culture flasks with serum-containing medium for 20 min, allowing for fibroblast attachment. Unattached cells, i.e., cardiomyocytes, were moved to gelatin-/fibronectin-coated 6-well culture plates (3.75 × 10^5^/mL density). Fibroblasts were cultured for one week, split, and seeded on 6-well culture plates (1.8 × 10^5^/mL density). Cells were kept in a 37 °C, 5% CO_2_ humidified incubator. We have previously confirmed the purity of our cultures, including 800-fold higher cardiac troponin I (TnnI) mRNA in cardiomyocytes vs. fibroblasts [20]. N = 3 technical replicates from *n* = 2 (mRNA) or *n* = 3–4 (protein) separate cell isolations were used.

Cultured rat cardiomyocytes and fibroblasts were transduced with a human adenovirus 5 encoding mouse Sdc4 controlled by the CMV promoter (Ad-mSDC4, ADV-271493, Vector Biolabs, Malvern, PA, USA) or empty control vector (Ad-CMV-Null, #1300, Vector Biolabs), as described [19]. There is 94% sequence identity between rat (P34901) and mouse (U35988) syndecan-4 protein. mRNA analysis allowed for specific quantification of the overexpressed vs. endogenous *Sdc4* RNA since mouse syndecan-4 was overexpressed in rat cells. Virus titer was 5 × 10^6^ plaque forming units (PFU)/mL medium. Cells were serum-starved for 24 h prior to treatment with pro-inflammatory cytokines and their inhibitors: recombinant rat IL-1β (501-RL, R&D System, Minneapolis, MN, USA), recombinant rat TNFα (510-RT, R&D Systems), recombinant human IL-1 receptor antagonist (IL-1Ra; SRP3084, Sigma-Aldrich, St. Louis, MO, USA) and anti-TNFα antibody (infliximab; Inflectra, Pfizer). Cells were harvested after 24 h and mRNA and protein stored at −70 °C.

### 2.4. ELISA Analyses of Serum Samples

We detected human syndecan-4 in serum using a commercially available human syndecan-4 ELISA kit (27188, IBL, Gunma, Japan) according to protocol, by one experienced researcher blinded to patient group. The two specific antibodies used in this ELISA were generated against residues 27–39 and residues 19-145 of the recombinant syndecan-4 ectodomain fused to the maltose binding protein [29]. The detection limit is 20 pg/mL.

A literature search was conducted in PubMed, National Institutes of Health (NIH), with the search words “syndecan-4” and “biomarker”. Abstracts of the resulting hits (*n* = 91) were read to identify those reporting ELISA data of syndecan-4 from human blood samples. The next and final selection criteria were papers specifying the use of serum samples, the IBL human syndecan-4 ELISA, and including a control group of healthy subjects. These data were used to generate a table overviewing absolute levels of serum syndecan-4 in various cohorts, to compare to our data. 

### 2.5. RNA Isolation and Quantitative Real-Time PCR (qRT-PCR)

Total RNA was isolated from cells using RNeasy mini (74106, Qiagen Nordic, Oslo, Norway), as previously described [20]. In brief, reverse transcription and cDNA synthesis were performed with the iScript cDNA Synthesis Kit (Bio-Rad Laboratories, Inc., Hercules, CA, USA). Pre-designed TaqMan assays (Rn00561900_m1 and Rn06291926_g1) were used with TaqMan Universal PCR Master Mix (#4304437, both from Applied Biosystems, Foster City, CA, USA) to quantify syndecan-4 mRNA levels by qRT-PCR. Results were detected on the QuantStudio 3 Real-Time PCR System (Applied Biosystems). mRNA data were normalized to expression of ribosomal protein L4 (*Rpl4*).

### 2.6. Protein Isolation and Immunoblotting

Myocardial tissue and whole-cell protein lysates were prepared as previously described [20], using a PBS lysis buffer containing 1% Triton X-100 (Sigma, MI), 0.1% Tween-20 (Sigma), 0.1% sodium dodecyl sulfate (SDS) and protease inhibitors (Complete EDTA-free tablets, Roche Diagnostics, Oslo, Norway). Supernatants were collected after centrifugation at 20,000× *g* for 10 min at 4 °C and stored at −70 °C. Protein concentrations were calculated after running the Micro BCA kit (#23235, Thermo Fisher Scientific, Waltham, MA). SDS-PAGE and immunoblotting were run in line with the Criterion BIO-RAD protocol, as described [38]. Proteins were separated by SDS-PAGE using Criterion 4–15% TGX gels (Cat# 5671084, Bio-Rad) and transferred to polyvinylidene difluoride membranes (#1704156, Bio-Rad) running the Trans-Blot Turbo Transfer System. Blots were blocked in casein (Roche Diagnostics, Oslo, Norway) or non-fat dry milk (Sigma) and incubated with primary antibodies in blocking solution overnight. HRP-conjugated secondary antibodies (1:1750, Amersham/GE HealthCare, Buckinghamshire, UK) were applied to all blots. Blots were developed using ECL Prime (# RPN2232, GE Healthcare, Chicago, IL, USA) in the LAS 4000 (Fujifilm, Tokyo, Japan). Processing was performed in Adobe Photoshop 2023 and quantification in ImageJ v1.52a (NIH). Protein levels were normalized to loading controls. The primary antibodies used were anti-syndecan-4, detecting an epitope in the short, highly conserved 28 amino acid cytoplasmic domain (1:1000, custom made from Genscript Corporation, Piscataway, NJ, USA; see [16] for details including epitope mapping), and anti-vinculin (1:1,000,000, V9131; Sigma-Aldrich) for loading control. 

### 2.7. Statistics

Data are expressed as dot plots with bar group means ± S.D. Statistical analyses were performed using GraphPad Prism 9 and statistical differences considered significant for *p* < 0.05. Significance was determined using Student’s paired and unpaired *t*-test or one-way ANOVA followed by Tukey’s post hoc test, as detailed in the figure and table legends.

## 3. Results

### 3.1. Syndecan-4 Is Shed in and from the Human Heart

To measure syndecan-4 shedding from cells or tissue and in blood, we performed Western blotting and ELISA, respectively (Figure 1A). The custom-made syndecan-4 antibody against the cytoplasmic tail of syndecan-4 (Figure 1A, left panel) detects both full-length (FL) syndecan-4 and the cellular fragment (CF) left in the cell after release of the ectodomain in mouse, rat and human samples (see [16,20] for details). Thus, the levels of the CF reflect syndecan-4 shedding were measured in tissue and cells. The detection of syndecan-4 FL (35–40 kDa) and the CF (10–15 kDa) in diseased human myocardium is shown here for demonstration of the protein bands and the ongoing syndecan-4 shedding observed in heart disease (Figure 1B), with biological variation in human samples, i.e., a stronger syndecan-4 signal in two out of the three samples tested. To measure the levels of the syndecan-4 shed ectodomain (SE) in blood samples, we used an ELISA kit equipped with an antibody against the ectodomain (Figure 1A, right panel). To test whether circulating syndecan-4 originated from the heart, we used ELISA to measure the venous-arterial difference in serum from the coronary sinus, i.e., venous blood from the heart, and the radial artery, of aortic stenosis patients during open heart surgery. Consistent with syndecan-4 ectodomains being released into the circulation from the human heart, syndecan-4 levels were higher in the coronary sinus (10.73 ± 4.24 vs. 6.50 ± 2.66 ng/mL in the radial artery) (Figure 1C). To study syndecan-4 shedding from major cell types in the heart, we prepared primary cardiomyocyte and cardiac fibroblast (Figure 1D) cultures from neonatal rats. To provide a robust assay for the measurement of syndecan-4 shedding in cardiac cells, cells were transduced with a virus encoding mouse syndecan-4 (AdSdc4) or vehicle (AdNull). Overexpression of syndecan-4 resulted in the expected increase in FL (at or somewhat below 37 kDa in fibroblasts vs. cardiomyocytes, respectively) and CF, the latter by constitutive shedding from cardiomyocytes (Figure 1E) and fibroblasts (Figure 1F). We considered our methods suited to detect syndecan-4 shedding from cardiac cells, in myocardial tissue, and in blood from patients.

### 3.2. Serum Syndecan-4 Levels Do Not Reflect the Inflammatory Status of the Heart in Patients with Heart Disease

To test our hypothesis that syndecan-4 could constitute a blood biomarker of cardiac inflammation, syndecan-4 was measured by ELISA in serum samples from a total of 549 well-characterized patients with cardiomyopathies of various etiologies and stages of disease, or acute MI. Flow diagrams of the patient selection process for the two cohorts are shown in Figure 2. The patient characteristics were previously reported [33,34,35]. 

To investigate whether circulating syndecan-4 levels were elevated in patients with a high degree of inflammation in the myocardium, we separated the DCM patients into inflammatory (chronic myocarditis, *n* = 71) and non-inflammatory (*n* = 318) DCM, where inflammation was defined as >14 infiltrating cells/mm^2^ on immunohistochemistry. However, we found no differences in serum syndecan-4 levels between the two groups (average 16.26 ng/mL vs. 15.20 ng/mL in inflammatory and non-inflammatory DCM, respectively) (Figure 3). 

To examine the possibility that syndecan-4 levels could differentiate between subgroups of cardiomyopathies characterized by acute immune responses, we sub-divided patients based on their specific diagnoses, i.e., acute myocarditis (*n* = 15), acute pericarditis (*n* = 3) and acute perimyocarditis (*n* = 23). Serum syndecan-4 was on average 13.66 ng/mL in myocarditis, 13.54 ng/mL in pericarditis and 17.47 ng/mL in perimyocarditis, and again, no differences were observed between the groups (Figure 3). Apart from the very small group of pericarditis patients (*n* = 3), every patient group analyzed so far displayed a comparable variation of syndecan-4 levels, with comparable minimum (min.; 2.44–6.95 ng/mL) and maximum (max; 31.28–38.72 ng/mL) levels between the groups, and an average range of 30.90 ng/mL within each group. Thus, our results suggest that circulating syndecan-4 is of limited value in the clinic with regard to reflecting cardiac inflammation. 

Acute MI is another condition with a high degree of inflammation. We obtained serum samples at hospital admission (day 0), and at day 3 and day 30 post-MI from each patient (*n* = 119). Syndecan-4 showed slightly higher mean values at day 3 (16.29 ng/mL, *p* = 0.0002) and day 30 (15.78, *p* = 0.013) when paired statistical comparisons were made to levels at day 0 (14.39 ng/mL) in each patient (Figure 3). Thus, patients with acute MI display higher serum levels of syndecan-4 during the inflammatory phase post-MI. Of note, however, the mean and range of syndecan-4 levels (min. values of 0.86, 5.84 and 5.94 ng/mL and max. values of 32.20, 32.24 and 34.80 ng/mL at day 0, 3 and 30, respectively) in post-MI patients did not differ from the cardiomyopathy groups analyzed.

Lastly, after demonstrating that circulating syndecan-4 levels did not differentiate between the cardiac pathologies investigated, we wondered whether syndecan-4 levels were elevated in patients vs. healthy controls. On our end, we had access only to a small number of control subjects (*n* = 5), and found that these showed an average level of serum syndecan-4 of 13.30 ng/mL (Figure 3). The range of syndecan-4 values (min. value of 7.88 and max. value of 21.09 ng/mL) in healthy controls overlapped with patients of all etiologies, apart from the pericarditis group. Thus, we were unable to conclude whether serum syndecan-4 was increased vs. the controls in our cardiomyopathy or acute MI patient cohorts.

### 3.3. Interleukin 1β-Increased Syndecan-4 Levels Is Attenuated by Immunomodulatory Therapy in Cultured Cardiomyocytes

We capitalized on our previous finding that mediators of innate immunity, i.e., TNFα and IL-1β, upregulate production and shedding of syndecan-4 from cardiac myocytes and fibroblasts [18,20], in order to investigate whether immunomodulatory therapy affected syndecan-4 levels in such cells. Primary neonatal rat cardiomyocytes were treated with recombinant IL-1β and TNFα of rat origin. Treatment with IL-1β resulted in a 3.4-fold increase in rat syndecan-4 mRNA (Figure 4A). Interestingly, we found that co-treatment with the immunomodulating IL-1 receptor antagonist IL-1Ra resulted in an attenuated syndecan-4 increase. Consistent with the upregulated syndecan-4 mRNA, immunoblots showed that IL-1β increased syndecan-4 FL and CF 3.8- and 2.5-fold vs. the non-treated, respectively, in cardiomyocytes overexpressing mouse syndecan-4 (Figure 4B). Upon co-treatment with IL-1Ra, the FL levels were not increased vs. the non-treated; however, the levels of CF appeared not to be affected. When performing a separate statistical analysis comparing FL and CF levels in IL-1β vs. IL-1β + IL-1Ra only, IL-1Ra resulted in a 25% reduction in FL (*p* < 0.05) and a tendency for reduced CF (*p* = 0.07, 15%). 

Treatment with TNFα resulted in 2-fold higher levels of endogenous rat syndecan-4 mRNA (Figure 4C). However, co-treatment with its inhibitor, the anti-TNFα antibody infliximab, did not affect syndecan-4 production. Similarly, treatment with TNFα resulted in the upregulation of syndecan-4 with 2.9- and 3.1-fold increases in FL and CF protein, respectively, in cardiomyocytes overexpressing mouse syndecan-4 (Figure 4D), and co-treatment with infliximab did not affect TNF-α-induced syndecan-4 protein levels, whether compared to non-treated or in a separate statistical analysis comparing FL and CF levels in TNFα vs. TNFα + infliximab only.

### 3.4. Interleukin 1β-Increased Syndecan-4 Levels and Shedding Is Attenuated by Immunomodulatory Therapy in Cultured Cardiac Fibroblasts

Primary neonatal rat cardiac fibroblasts were treated with recombinant IL-1β and TNFα of rat origin. Treatment with IL-1β resulted in a 7.7-fold increase in syndecan-4 mRNA (Figure 5A). Importantly, co-treatment with IL-1Ra resulted in a 50% reduction in IL-1β-induced syndecan-4 production. These effects of IL-1β and IL-1Ra were also evident when we investigated syndecan-4 protein levels. IL-1β induced a 20.2-fold increase in syndecan-4 FL and a 14.8-fold increase in CF (Figure 5B). Co-treatment with IL-1Ra resulted in a clear reduction of IL-1β-induced syndecan-4 FL protein, while CF levels showed no change. When performing a separate statistical analysis comparing FL and CF levels in IL-1β vs. IL-1β + IL-1Ra only, IL-1Ra resulted in a 39% reduction in FL (*p* < 0.001) and a 27% reduction in CF (*p* < 0.01). 

Treatment with TNFα resulted in a 2.4-fold increase in syndecan-4 mRNA (Figure 5C). Interestingly, the TNFα-induced increase was attenuated when the cardiac fibroblasts were co-treated with infliximab. In line, TNFα resulted in a 2.4- and 4.3-fold increase in syndecan-4 FL and CF protein, respectively, in cardiac fibroblasts overexpressing mouse syndecan-4 (Figure 5D). However, infliximab resulted in a non-significant tendency for increased syndecan-4 FL and CF protein levels, and consequently, did not have an attenuating effect on TNF-α-induced syndecan-4 protein levels when compared to non-treated or in the separate statistical analysis of TNFα vs. TNFα + infliximab. 

Taken together, these results from primary heart cultures showed that IL-1β was a strong inducer of syndecan-4 mRNA and protein production in cardiac myocytes and fibroblasts, and that treatment with IL-1Ra reduced the levels of FL syndecan-4 and its shedding. Although TNFα was a potent inducer of syndecan-4 mRNA and protein in cardiac myocytes and fibroblasts, co-treatment with infliximab did not alter the FL or shed protein. 

## 4. Discussion

The present work investigates (i) whether circulating levels of syndecan-4 reflect the cardiac inflammatory status in patients with heart disease, and (ii) the effects of immunomodulatory inhibitors used in clinical practice on syndecan-4 levels and shedding in vitro. The presence of syndecan-4 shedding fragments in diseased myocardial tissue, and the increased veno-arterial difference of shed syndecan-4 ectodomains across the heart, indicated that syndecan-4 ectodomains are released in and from the human heart. Importantly, serum syndecan-4 showed comparable levels in patients with cardiomyopathies of different etiologies, i.e., inflammatory vs. non-inflammatory DCM, myocarditis, pericarditis and perimyocarditis, and following MI, indicating that syndecan-4 has limited value as a clinical circulating biomarker of cardiac inflammation. In cardiac cells in vitro, a decrease in IL-1β-induced syndecan-4 mRNA, protein and shedding following treatment with IL-1Ra suggested that syndecan-4 production and shedding is among the downstream targets of IL-1 blockage. Treatment with an anti-TNFα antibody, infliximab, failed to show a similar effect on the TNFα-induced upregulation of syndecan-4 protein levels and shedding. 

Published studies have indicated that circulating syndecan-4 holds promise as a cardiac biomarker. Although these include cohorts with relatively few patients, circulating levels of syndecan-4 are increased after acute MI [29], resistant hypertension [39] and chronic heart failure [30]. Indicating that syndecan-4 could be a blood biomarker reflecting cardiac remodeling and function, serum syndecan-4 correlated positively with LV geometry and negatively with pumping function as measured by EF [30,31]. Based on the knowledge that syndecan-4 FL levels and the shedding of its ectodomain are upregulated in response to innate immunity pathways [18,20], we set out to evaluate whether syndecan-4 could also hold promise as a circulating biomarker of cardiac inflammation. We tested this by measuring the serum concentrations of syndecan-4 in patients with DCM of inflammatory and non-inflammatory origin, patients with acute cardiomyopathies, and in patients with acute MI, when admitted and after three and 30 days. However, our results from a high number of patients did not show differences in syndecan-4 levels among patients with chronic, acute or minor inflammation. Overall, we conclude from our findings that there is limited value in the use of syndecan-4 as a clinical circulating biomarker of cardiac inflammation. Despite our disappointing results, we believe there is scientific value in publishing such neutral data. We performed the analyses on two well-characterized patient cohorts that included a high number of patients. Respect for the patients and the valuable samples they provide to research, together with the cost of analyses, certainly justify that information of this scale ought to be shared among researchers to avoid unfruitful replications and to guide future hypotheses. 

Interestingly, we did demonstrate that serum syndecan-4 increased during the inflammatory phase in patients with acute MI, with levels peaking at day 3 and remaining elevated at day 30, compared to at hospital admission. These results are supportive of a role for syndecan-4 shedding in the ischemic heart, and are in line with previous studies showing increased levels of circulating syndecan-4 after MI [29,32]. Such a role has also been demonstrated for syndecan-1, which is shed during ischemic conditions of the heart causing endothelial damage, e.g., cardiac arrest [22,23], chronic heart failure [24] and microvascular occlusions in ST elevation MI [40]. 

Another aspect of our study of serum syndecan-4 in patients is its level in healthy controls. Our study failed to conclude whether circulating syndecan-4 levels are at all increased in heart disease patients vs. healthy controls. In our study, the average level in healthy controls was 13.3 ng/mL. Considering that the mean and range of syndecan-4 levels in our patients overlapped with levels in healthy controls, albeit the latter in a low number, we evaluated the levels of serum syndecan-4 obtained using the same ELISA kit and sample material in published, historical controls (Table 1). Although a number of studies have found significant changes in serum syndecan-4 in patient cohorts vs. healthy subjects, Table 1 demonstrates that there is a spread in the reported mean values for circulating syndecan-4 in control populations, ranging from 4.34 to 18.99 ng/mL [30,39,41,42,43,44,45]. Of note, syndecan-4 levels measured in healthy controls in one study are comparable to disease states in others. Collectively, these studies are in line with our current results showing overlapping syndecan-4 concentrations among patient groups. Consequently, to advance our understanding of shed syndecan-4 as a biomarker in pathology, efforts should be directed towards truly defining circulating syndecan-4 levels in healthy controls.

Although we did not find that circulating syndecan-4 levels reflected the inflammatory status of the heart, we found syndecan-4 shedding fragments within the diseased human myocardial tissue and increased veno-arterial difference of shed syndecan-4 ectodomains across the heart, showing that syndecan-4 ectodomains were released in and from the human heart. Using a novel approach with syndecan-4 overexpression, IL-1β and TNFα induced a robust increase in syndecan-4 production in cardiac myocytes and fibroblasts in cultured, in line with our previous findings on endogenous syndecan-4 [20]. Thus, the cardiac increase was induced by activation of innate immunity signaling. 

We have shown that when cardiomyocytes and cardiac fibroblasts are exposed to shed ectodomains of syndecan-4, they upregulate key molecules involved in inflammatory responses, i.e., cytokines and cell adhesion receptors involved in the binding of immune cells [18]. Thus, we believe that the increased release of syndecan-4 ectodomains resulting from augmented shedding in diseased hearts generates soluble pro-inflammatory molecules that activate immune cell recruitment pathways and propagate immune responses locally in the cardiac tissue. 

We and others have demonstrated that syndecan-4 has pro-hypertrophic effects in cardiomyocytes and pro-fibrotic effects in cardiac fibroblasts [14,16,17,46,47,48]. This means that treatments reducing syndecan-4 levels could be anti-hypertrophic and anti-fibrotic, as well as anti-inflammatory. After discovering here, and in our previous study [20], that IL-1β is a potent driver of syndecan-4 production, we investigated whether immunomodulatory agents used in current clinical practice could modulate cardiac syndecan-4 levels. To this end, we found a decrease in IL-1β-induced syndecan-4 mRNA, protein and shedding following treatment with IL-1Ra (also called anakinra) in vitro, suggesting that syndecan-4 production and shedding is among the downstream targets of IL-1 blockage. Interestingly, IL-1Ra is an IL-1 inhibitor currently under investigation as therapy after acute MI and chronic heart failure [49]. Similar to other therapies targeting pro-inflammatory cytokines, studies on the effect of IL-1Ra have reported conflicting and oftentimes disappointing results. However, there have been several clinical trials showing promising effects of IL-1 blockage by IL-1Ra with regard to peak oxygen consumption and heart failure outcomes [7,50]. The exploration of immunomodulating therapies in patients with cardiac disease remains a field of continued research efforts. Our results, showing that treatment with IL-1Ra reduced the upregulation and shedding of syndecan-4 from cardiac cells, indicate that syndecan-4 levels will be reduced in hearts of patients receiving immunosuppressive therapies, with likely downstream consequences. 

## 5. Conclusions

In conclusion, increased syndecan-4 production and the release of its ectodomain take place during the innate immune response of the heart, yet levels of syndecan-4 in the circulation were not different between patients with inflammatory and non-inflammatory heart disease. However, we did observe increased circulating levels 3–30 days post-MI compared to at hospital admission, supporting a role for syndecan-4 shedding in the ischemic heart. Thus, we conclude that syndecan-4 had limited clinical value as a circulating biomarker of cardiac inflammation and that immunomodulating therapies targeting IL-1β likely will affect the molecular processes that shed syndecan-4 ectodomains play locally in cardiac tissue. 

## Figures and Tables

**Figure 1 biomedicines-11-01066-f001:**
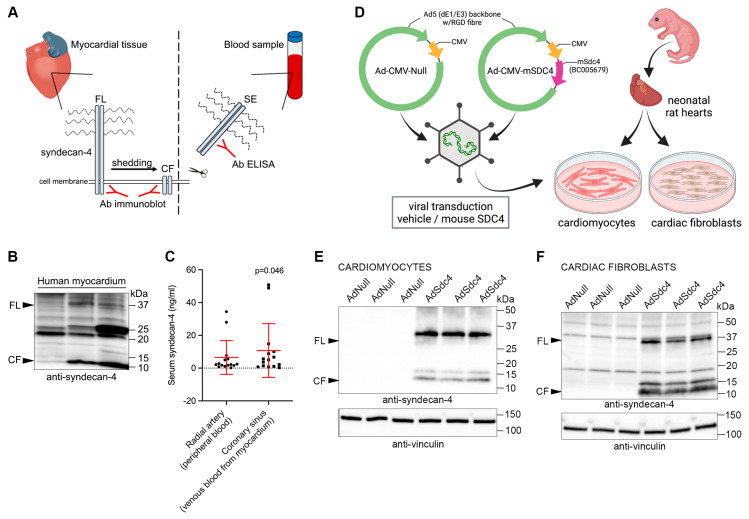
Syndecan-4 is shed in and from the human heart. (**A**) Schematic illustrating the antibody used to detect full-length (FL) syndecan-4 and the cellular fragment (CF) left in cells after shedding of the ectodomain, by immunoblotting of myocardial tissue or cellular protein lysates (left panel), and the antibody detecting the shed ectodomain (SE) in blood (right panel) by enzyme-linked immunosorbent assay (ELISA). (**B**) Representative immunoblot of human FL (upper arrowhead) syndecan-4 and the CF (lower arrowhead) in myectomy septum biopsies from patients with hypertrophic obstructive cardiomyopathy, *n* = 3. (**C**) Serum syndecan-4 levels (ng/mL) in peripheral blood from the radial artery and blood from the coronary sinus of aortic stenosis patients obtained during open heart surgery, *n* = 15. The numerical data supporting the graph can be found in Appendix A. (**D**) Schematic illustrating the adenoviral vectors used to transduce cultures of primary neonatal rat cardiac cells. Cells were transduced with an Adenovirus serotype 5 (dE1/E3) with RGD fiber modification encoding mouse syndecan-4 (AdSdc4) or vehicle (AdNull). (**E**,**F**) Representative immunoblots of mouse syndecan-4 FL and CF in primary cultures of neonatal rat cardiomyocytes (*n* = 9, (**E**)) and cardiac fibroblasts (*n* = 12, (**F**)). Vinculin was used for loading control. Data are presented as mean ± S.D. Statistical differences were tested using Student’s paired *t*-test (**C**).

**Figure 2 biomedicines-11-01066-f002:**
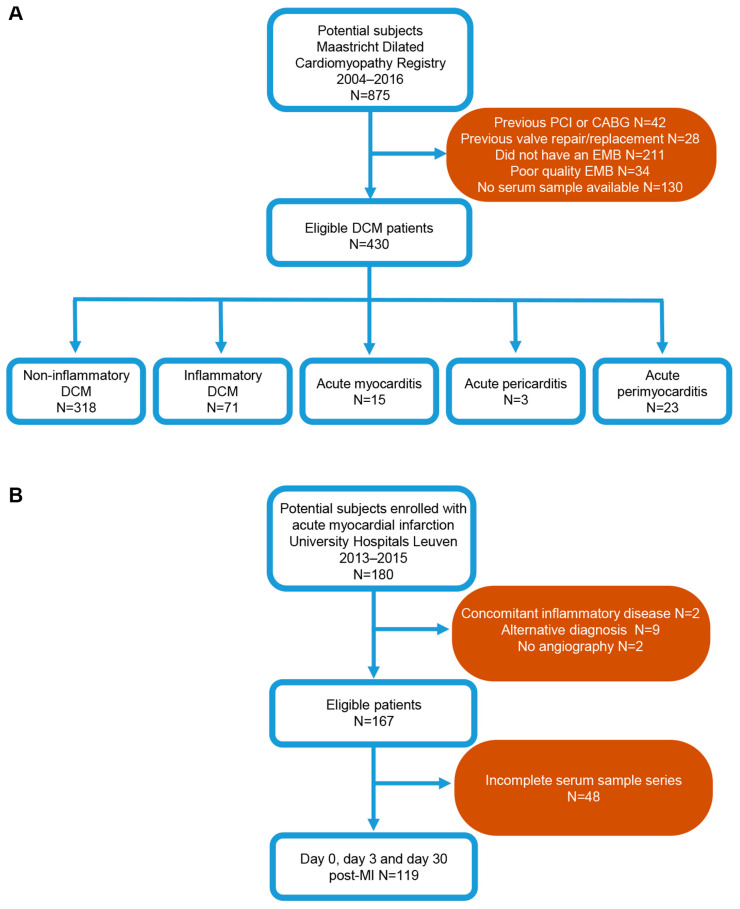
Flow diagrams of included patients. (**A**) Flow diagram of cardiomyopathy patients included from the Maastricht Dilated Cardiomyopathy Registry. (**B**) Flow diagram of patients with acute myocardial infarction (MI) enrolled at University Hospitals Leuven.

**Figure 3 biomedicines-11-01066-f003:**
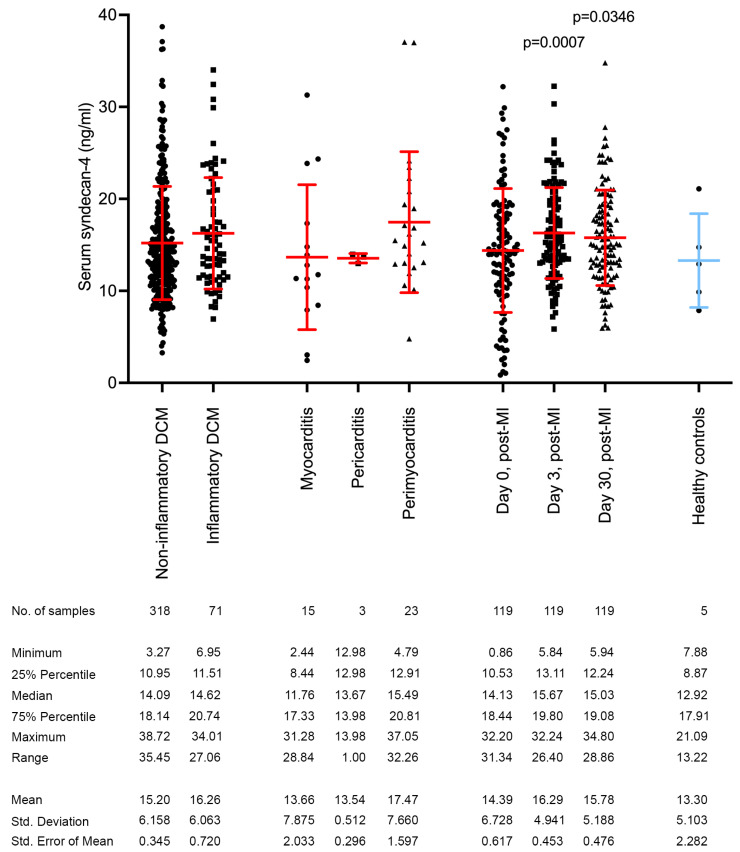
Serum syndecan-4 levels do not reflect the inflammatory status of the heart in patients with heart disease. Serum samples were collected from patients with cardiomyopathy of various etiologies. Patients were extensively characterized. Serum syndecan-4 levels (ng/mL) were measured by enzyme-linked immunosorbent assay (ELISA) by one experienced researcher blinded to patient characteristics. The cohort (*n* = 430) consisted of subgroups with specific diagnoses, i.e., inflammatory (*n* = 71) and non-inflammatory (*n* = 318) dilated cardiomyopathy (DCM), myocarditis (*n* = 15), pericarditis (*n* = 3) and perimyocarditis (*n* = 23). Serum samples were obtained from *n* = 119 patients with acute myocardial infarction (MI), with or without ST-elevation, at day of admission (day 0), day 3 day and day 30 post-MI. We included serum samples from *n* = 5 healthy controls. Data in the graph are presented as individual data points and mean ± S.D for patients (red) and controls (blue). The numerical data supporting the graph can be found in Appendix A. Descriptive statistics are included underneath the graph. Statistical differences were tested using ordinary or repeated measures one-way ANOVA with Tukey’s post hoc test (ordinary: myocarditis, pericarditis and perimyocarditis; repeated measures: post-MI cohort) or Student’s unpaired t-test (inflammatory vs. non-inflammatory DCM).

**Figure 4 biomedicines-11-01066-f004:**
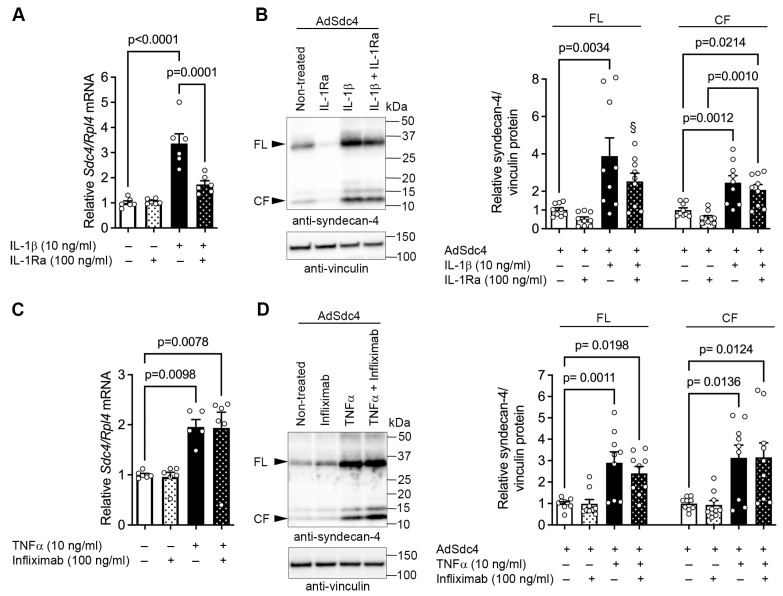
Attenuation of interleukin 1β-increased syndecan-4 levels immunomodulatory therapy in cultured cardiomyocytes. Primary cardiomyocyte cultures were prepared from neonatal rat hearts. (**A**) Relative mRNA expression of endogenous rat syndecan-4 (Sdc4) in cardiomyocytes following 24 h treatment with interleukin (IL)-1β (10 ng/mL) compared to non-treated control or co-treatment with the IL-1 receptor antagonist (IL-1Ra; 100 ng/mL), *n* = 6. (**B**) Representative immunoblot and quantification of full-length (FL) syndecan-4 and the cellular fragment (CF) seen in cells after release of the ectodomain, in cardiomyocytes transduced with an adenovirus encoding mouse syndecan-4 (AdSdc4) and after 24 h treatment with IL-1β compared to non-treated or co-treatment with IL-1Ra, *n* = 9. (**C**) Relative mRNA levels of endogenous rat syndecan-4 in cardiomyocytes after 24 h treatment with tumor necrosis factor (TNF)α (10 ng/mL) compared to non-treated control or co-treatment with the anti-TNFα antibody infliximab (100 ng/mL), *n* = 6. (**D**) Representative immunoblot and quantification of syndecan-4 FL and CF in cardiomyocytes transduced with mouse AdSdc4 and treated for 24 h with TNFα vs. non-treated or co-treatment with infliximab, *n* = 9. mRNA data were normalized to ribosomal protein L4 (Rpl4). Vinculin was used for loading control. Data are presented as mean ± S.D. Statistical differences were tested using one-way ANOVA with Tukey’s post hoc test ((**A**–**D**), *p*-values given) or Student’s unpaired *t*-test ((**B**) § *p* < 0.05, comparing Il-1Ra to IL-1β alone, not relative to control). The numerical data supporting the graphs can be found in Appendix A.

**Figure 5 biomedicines-11-01066-f005:**
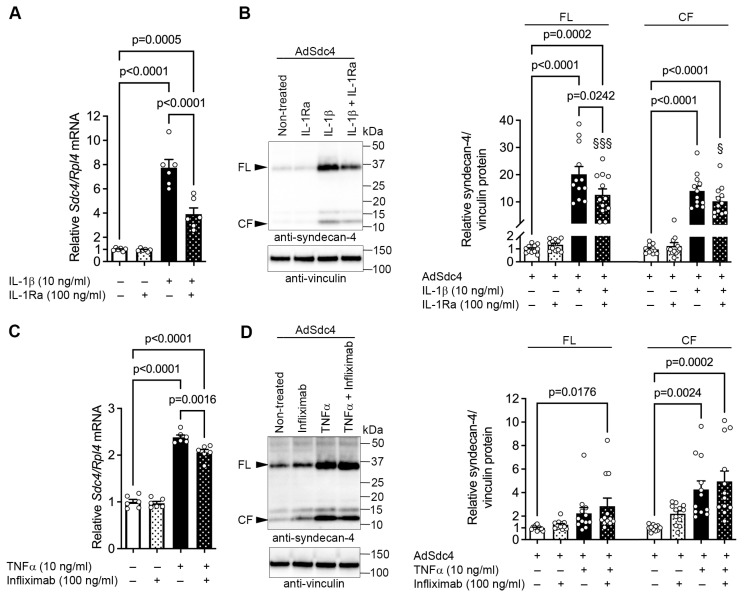
Interleukin 1β-increased syndecan-4 levels and shedding are attenuated by immunomodulatory therapy in cultured cardiac fibroblasts. Primary cardiac fibroblast cultures were prepared form neonatal rat hearts. (**A**) Relative mRNA levels of rat syndecan-4 (Sdc4) in cardiac fibroblasts after 24 h treatment with interleukin (IL)-1β (10 ng/mL) compared to non-treated control or co-treatment with the IL-1 receptor antagonist (IL-1Ra; 100 ng/mL), *n* = 6. (**B**) Representative immunoblot and quantification of full-length (FL) syndecan-4 and the cellular fragment (CF) remaining after shedding of the ectodomain, in rat cardiac fibroblasts transduced with an adenovirus encoding mouse syndecan-4 (AdSdc4) and following 24 h treatment with IL-1β compared to non-treated or co-treatment with IL-1Ra, *n* = 12. (**C**) Relative mRNA of rat syndecan-4 in cardiac fibroblasts after 24 h treatment with tumor necrosis factor (TNF)α (10 ng/mL) vs. non-treated control or co-treatment with the anti-TNFα antibody infliximab (100 ng/mL), *n* = 6. (**D**) Representative immunoblot and quantification of syndecan-4 FL and CF in cardiac fibroblasts transduced with mouse AdSdc4 and treated for 24 h with TNFα vs. non-treated or co-treatment with infliximab, *n* = 12. mRNA data were normalized to ribosomal protein L4 (Rpl4). Vinculin was used for loading control. All data are presented as mean ± S.D. Statistical differences were tested using one-way ANOVA with Tukey’s post hoc test ((**A**–**D**), *p*-values given) or Student’s unpaired *t*-test ((**B**) § *p* < 0.05; §§§ *p* < 0.001, comparing Il-1Ra to IL-1β alone, not relative to controls). The numerical data supporting the graphs can be found in Appendix A.

**Table 1 biomedicines-11-01066-t001:** Historical syndecan-4 levels in serum from patients and healthy controls.

Human Syndecan-4 ELISA Kit—IBL 27188						
Healthy Controls		Patient Population				
N	Serum Syndecan-4 (ng/mL)	Disease	N	Serum Syndecan-4 (ng/mL)	*p*-Value vs. Healthy	Reference
21	5.7 ± 3.3	Chronic heart failure	45	22.5 ± 12.3	*p* < 0.01	[30]
11	15.1 ± 2.6	Acute pneumonia	30	24.7 ± 9.2	*p* = 0.006	[41]
45	16.05 ± 0.77	Idiopathic interstitial pneumonia	62	25.22 ± 3.72		[42]
		SD-IIP/AE-IIP	56	10.65 ± 0.73	*p* < 0.05 (SD-IIP)	
30	14.30 ± 5.34	Community-acquired pneumonia	103	9.54 ± 5.92		[43]
		SCAP/non-SCAP	149	10.15 ± 4.37	*p* < 0.001	
35	14.7 ± 0.6	Resistant hypertension	19	35.8 ± 5.3	*p* < 0.001	[39]
16	18.99 ± 1.36	Osteoarthritis	29	19,23 ± 9.16	NS	[44]
376	4.34 ± 1.78	Non-alcoholic fatty liver disease	157	11.25 ± 5.15	*p* < 0.001	[45]

Historical, published levels of serum syndecan-4 (ng/mL) in patient cohorts and healthy controls measured using the same ELISA kit (IBL 27188, Gunma, Japan) as in the present study. Our color-coding: blue, syndecan-4 in healthy controls; green, syndecan-4 increased in patients vs. healthy controls in given study; red, syndecan-4 reduced in patients vs. healthy controls; grey, no difference patients vs. healthy controls. SD-IIP, stable disease idiopathic interstitial pneumonia; AE-IIP, acute exacerbation-IIP, SCAP, severe community-acquired pneumonia (CAP); NS, non-significant. Data are presented as mean ± S.E.M [39,41,42,43,44] or mean ± S.D. [30,45].

## Data Availability

The data presented in this study are contained within this article and are available in the Appendix A.

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
