# Peer review of "Inflammation and Syndecan-4 Shedding from Cardiac Cells in Ischemic and Non-Ischemic Heart Disease"

_biomedicines, 2023, doi:10.3390/biomedicines11041066_

Round 1

Reviewer 1 Report

This is a well designed and written study. THe obtained data are weak presented and of significant value regrading the role of SDCs/4 in ischemic conditions. Authors could improve intro and/or discussion including relevant recent articles on the roles of SDCs and their shedding.

Author Response

Reviewer 1:

  • This is a well designed and written study. The obtained data are weak presented and of significant value regrading the role of SDCs/4 in ischemic conditions. Authors could improve intro and/or discussion including relevant recent articles on the roles of SDCs and their shedding.

We thank the reviewer for the positive and constructive comments. We have revised our manuscript to give our findings of increased circulating levels of syndecan-4 post-MI more emphasis and included references to more recent articles on the role of syndecan shedding in ischemic conditions.  (Introduction, lines 104-116; Discussion, lines 1120-1125; Conclusion, lines 1250-1257).

NB! The line numbers referred to correspond with the revised MS doc file with track changes.

Reviewer 2 Report

Dear Mari Strand and colleagues,

here are my comments on your manuscript reporting that the type I membran proten Syndecan-4 is no biomarker for inflammatory cardiac disorders.

Introduction: L54: please give the reader more background on the structure and functions of Syndecan-4. Please explain why the full-length protein runs at a higher MW as the expected ~20 kDa. Please remember to write for an educated lay audience :)

Results: L196-198: please explain why the mouse and not the human Syndecan-4 was expressed in the rat cells. Importantly, please be precice throughout the manuscript referring to murine Syndecan-4 or human Syndecan-4 where the corresponding proteins are studied. Currently, readers have the impression that the same Syndecan-4 protein has been analysed, which is not the case.

Figure 1B: please comment in the main text why there is no Syndecan-4 signals in the first (left) lane. Please refer to murine and rat species in the figure legend were appropriate.

L215: please introduce MI

L256: I am not sure whether the term "significantly" is correct looking at the high base rate variation of Syndecan-4 levels in healthy people and patients. Please indicate the alpha value of the statistical analysis. I think a phrase like " syndecan-4 protein levels showed a tendency to a slightly higher mean value" would be more approriate.

Section 3.3. L289: Please remind the reader here why IL-1beta and TNF-alpha are investigated here. Please indicate the source of IL-1beta and TNF-alpha (human or murine proteins?).

Figure 4B, D: please check the protein size markers as the fulllength Syndecan-4 appears to run below the 37kDa marker. If the murine Syndecan-4 runs differently from the human protein, please say so. Please refer to murine and rat sources in the figure legend.

Section 3.4: L: 342 please say that infliximab increases protein levels (Fig 5D). The term "not alter" is not correct. Please allert the reader as well to the fact that these studies were done with viral transfected cells.

Author Response

Reviewer 2:

Dear Mari Strand and colleagues, here are my comments on your manuscript reporting that the type I membran proten Syndecan-4 is no biomarker for inflammatory cardiac disorders.

  • Introduction: L54: please give the reader more background on the structure and functions of Syndecan-4. Please explain why the full-length protein runs at a higher MW as the expected ~20 kDa. Please remember to write for an educated lay audience :)

We thank the reviewer for the constructive and thorough feedback. In the revised manuscript, we have added more background on the structure and function of syndecan-4 in the Introduction, which we hope covers the existing literature and will engage the reader (Introduction, lines 104-116) including an explanation of the expected MW for syndecan-4 (lines 110-112).

NB! The line numbers referred to correspond with the revised MS doc file with track changes.

  • Results: L196-198: please explain why the mouse and not the human Syndecan-4 was expressed in the rat cells. Importantly, please be precice throughout the manuscript referring to murine Syndecan-4 or human Syndecan-4 where the corresponding proteins are studied. Currently, readers have the impression that the same Syndecan-4 protein has been analysed, which is not the case.

Thank you for pointing this out. We chose to express mouse syndecan-4, which shared high sequence homology (94% identity) to rat, in our primary rat cardiac cell cultures. This was done to easily separate between the overexpressed and endogenous protein on RNA level. We have added this information to the Methods section 2.3 (lines 242-245). We agree that we should be more precise in the labeling and reference to the source of syndecan-4 reported in each figure and have added this information in figure legends and text throughout the manuscript. To make the distinction clearer, we made a new panel in Figure 1 (Figure 1D) illustrating the specific vectors that were used to overexpress syndecan-4 in the primary rat cardiac cells.

  • Figure 1B: please comment in the main text why there is no Syndecan-4 signals in the first (left) lane. Please refer to murine and rat species in the figure legend were appropriate.

Thank you for addressing this potential confusion. We have commented in the text about biological variation in human cardiac biopsies, demonstrated by the stronger syndecan-4 signal in two out of three shown samples (results section 3.1, lines 714-715). Further, we have now carefully referred to the specific species in all the relevant figure legends, and hope this is now readily understandable for the readers.

  • L215: please introduce MI

We have introduced the term myocardial infarction (MI) in the abstract (Line 26) and the introduction (Line 118). L215 was part of the figure legend to Figure 1, where MI is not part of the figure, however, we believe that introducing MI in the legend of Figure 2 is necessary, and this has been done in the revised manuscript (Line 817).

  • L256: I am not sure whether the term "significantly" is correct looking at the high base rate variation of Syndecan-4 levels in healthy people and patients. Please indicate the alpha value of the statistical analysis. I think a phrase like " syndecan-4 protein levels showed a tendency to a slightly higher mean value" would be more approriate.

We certainly agree with the reviewer on this point and have rephrased this part of the Results section 3.2 (line 840).

  • Section 3.3. L289: Please remind the reader here why IL-1beta and TNF-alpha are investigated here. Please indicate the source of IL-1beta and TNF-alpha (human or murine proteins?).

We thank the Reviewer for this suggestion and have added a sentence in the Results section 3.3 to remind the reader why TNFa and IL-1b were used in these experiments (lines 884-888), including information about the source of the cytokines (line 888). This information is also found in the Methods section 2.3 (line 247).

  • Figure 4B, D: please check the protein size markers as the fulllength Syndecan-4 appears to run below the 37kDa marker. If the murine Syndecan-4 runs differently from the human protein, please say so. Please refer to murine and rat sources in the figure legend.

We thank the Reviewer for this thorough feedback. Indeed, the size markers are correctly placed. As is also shown in Figure 1E, the cardiomyocyte band appears somewhat below 37 kDa, which is a consistent finding (see also Figure 4). We have commented on this size difference in cardiomyocytes vs. cardiac fibroblasts in the Results section 3.1 (line 744). 

  • Section 3.4: L: 342 please say that infliximab increases protein levels (Fig 5D). The term "not alter" is not correct. Please allert the reader as well to the fact that these studies were done with viral transfected cells.

Indeed, the reviewer is correct in pointing out that the protein levels of syndecan-4 could be increased after infliximab treatment (ref. Fig. 5D). As this change is statistically non-significant, we concluded that infliximab did not alter TNFα- induced syndecan-4 protein levels. Despite the lack of significant effect, we agree with the reviewer that the term “not altered” is misleading, and as the increase in protein levels following infliximab treatment contrasts the findings from RNA, it is worth alerting the readers to this. We have changed the phrasing of our conclusion accordingly in the results section 3.4 (lines 950-952. We have specified in the results section 3.3 (line 902) and section 3.4 (lines 949-950), in addition to the figure legends, that the results from figure 4D and 5D are from cells overexpressing syndecan-4.

Reviewer 3 Report

In the original article 'Inflammation and syndecan-4 shedding from cardiac cells in ischemic and non-ischemic heart disease' submitted by Mari E. Strand et al. to Biomedicines, the authors investigate if soluble syndecan-4, cleavaged by shedding from cardiac cells is a potential biomarker for cardiac inflammation in different heart diseases. Although the results are partially dissapointing, the authors highlight in the discussion in a nice and very convincing way, that also neutral data deserve publication! And I absolutly agree to this statement! The authors are correct to underline that also neutral data are data which should be published. However, I have some points which can improve the quality of this manuscript and therefore I suggest a minor revision. I will explain my points in the following paragraphs:

1.) The abbreviation DCM should be introduced within the abstract (line 24).

2.) Could you indicate the relevance of infliximab also in the abstract?

3.) You write in the manuscript frequently 'Cardiac cells'. Do you mean cardiomycytes or should this be unspecific?

4.) It would be great to inidicate in line 46 that even genetic mouse models for modeling of genetic cardiomyopathies present inflammation. Please cite in this context the following paper:

Brodehl, A., Belke, D. D., Garnett, L., Martens, K., Abdelfatah, N., Rodriguez, M., ... & Gerull, B. (2017). Transgenic mice overexpressing desmocollin-2 (DSC2) develop cardiomyopathy associated with myocardial inflammation and fibrotic remodeling. PLoS One12(3), e0174019.

5.) Line 56 following: Is it known which enzyme is responsible for the shedding of syndecan-4? Is the cleavage site known? 

6.) Please indicate the details for analysis of inflammation (Line 84/85). Which antibody was used in which concentration. How was the counting done? etc.

7.) Do your DCM and HCM patients carry any genetic mutations? It would be interesting to indicate also in the introduction that 30-50 % of the DCM and/or HCM patients carry a genetic mutation in a relevant cardiomyopathy gene. In this context, it would be relevant to cite a review article describing the genetic background of cardiomyopathies. You can cite for example the following book chapter:

Gerull, Brenda, Sabine Klaassen, and Andreas Brodehl. "The genetic landscape of cardiomyopathies." Genetic Causes of Cardiac Disease (2019): 45-91.

8.) Could you please present vector maps in the supplements for your adenovirus 5 constructs?

9.)Line 161-171: Could you please indicate the used dilutions of the antibodies.

10.)Could you change the column plots to dot plots in figure 4 and figure 5?

11.) It would be great if the authors could present all data with SD instead of using S.E.M, since the reader is interested in the variation. Larger error bars are definitely not a problem.

Although neutral data are presented in this nice study, I still aggree with the authors that this kind of studies are necessary and should be published without any doubts! Therefore, I am highly convinced that the authors can fix the critized points and submit a good revised paper! Good luck with the revision!

Author Response

Reviewer 3:

In the original article 'Inflammation and syndecan-4 shedding from cardiac cells in ischemic and non-ischemic heart disease' submitted by Mari E. Strand et al. to Biomedicines, the authors investigate if soluble syndecan-4, cleavaged by shedding from cardiac cells is a potential biomarker for cardiac inflammation in different heart diseases. Although the results are partially dissapointing, the authors highlight in the discussion in a nice and very convincing way, that also neutral data deserve publication! And I absolutly agree to this statement! The authors are correct to underline that also neutral data are data which should be published. However, I have some points which can improve the quality of this manuscript and therefore I suggest a minor revision. I will explain my points in the following paragraphs:

We thank the reviewer for the constructive and thorough feedback, and for taking the time to review our paper.

NB! The line numbers referred to corresponds with the revised MS doc file with track changes.

  • The abbreviation DCM should be introduced within the abstract (line 24).

The abbreviation DCM has been added to the abstract (line 25).

  • Could you indicate the relevance of infliximab also in the abstract?

We have added more information about infliximab in the abstract (line 30).

  • You write in the manuscript frequently 'Cardiac cells'. Do you mean cardiomycytes or should this be unspecific?

Thank you for pointing this out. For clarity, we have specified cardiomyocytes or cardiac fibroblasts where appropriate throughout the manuscript.

  • It would be great to inidicate in line 46 that even genetic mouse models for modeling of genetic cardiomyopathies present inflammation. Please cite in this context the following paper:

Brodehl, A., Belke, D. D., Garnett, L., Martens, K., Abdelfatah, N., Rodriguez, M., ... & Gerull, B. (2017). Transgenic mice overexpressing desmocollin-2 (DSC2) develop cardiomyopathy associated with myocardial inflammation and fibrotic remodeling. PLoS One12(3), e0174019.

We thank the Reviewer for this input and have added this to the introduction (lines 95-96, citation number 5).

  • Line 56 following: Is it known which enzyme is responsible for the shedding of syndecan-4? Is the cleavage site known? 

We thank the Reviewer for bringing this up. In the revised manuscript, we have added information about syndecan-4 and its shedding, included the enzymes responsible, in the Introduction (lines 114-116).

  • Please indicate the details for analysis of inflammation (Line 84/85). Which antibody was used in which concentration. How was the counting done? etc.

We agree that some elaboration on the classification of inflammation vs. non-inflammation is appropriate and have added more information to the methods section 2.1 (lines 202-204).

  • Do your DCM and HCM patients carry any genetic mutations? It would be interesting to indicate also in the introduction that 30-50 % of the DCM and/or HCM patients carry a genetic mutation in a relevant cardiomyopathy gene. In this context, it would be relevant to cite a review article describing the genetic background of cardiomyopathies. You can cite for example the following book chapter:

Gerull, Brenda, Sabine Klaassen, and Andreas Brodehl. "The genetic landscape of cardiomyopathies." Genetic Causes of Cardiac Disease (2019): 45-91.

We thank the Reviewer for pointing out that also genetic cardiomyopathies, whether in patients or modelled in mice (point 4 above), show cardiac inflammation. Indeed, there were genetic mutations found in the patients constituting our material and the specific information is available in the references for the patient material in the Methods section 2.1 (lines 141-143) and 2.2 (line 222). In the revised manuscript Introduction, we have cited the suggested book chapter (line 96, citation number 4). 

  • Could you please present vector maps in the supplements for your adenovirus 5 constructs?

Thank you for this suggestion. We agree that vector maps showing our constructs for overexpression are appropriate and have added this in a new schematic in Figure 1D of the revised manuscript. In this new illustration we show the Sdc4 overexpression experiments performed in primary cultures of cardiac myocytes and fibroblasts from neonatal rats.

  • Line 161-171: Could you please indicate the used dilutions of the antibodies.

We have added the antibody dilutions to the methods section 2.3 (Lines 474, 479-480).

  • Could you change the column plots to dot plots in figure 4 and figure 5?

We have changed the column plots in figures 4 and 5 into dot plots.

  • It would be great if the authors could present all data with SD instead of using S.E.M, since the reader is interested in the variation. Larger error bars are definitely not a problem.

We have altered all figures to include SD instead of SEM.

Although neutral data are presented in this nice study, I still aggree with the authors that this kind of studies are necessary and should be published without any doubts! Therefore, I am highly convinced that the authors can fix the critized points and submit a good revised paper! Good luck with the revision!

Thank you!